# The Current Status of Neuroprotection in Congenital Heart Disease

**DOI:** 10.3390/children8121116

**Published:** 2021-12-02

**Authors:** Kei Kobayashi, Christopher Liu, Richard A. Jonas, Nobuyuki Ishibashi

**Affiliations:** 1Center for Neuroscience Research, Sheikh Zayed Institute for Pediatric Surgical Innovation, Children’s National Hospital, Washington, DC 20010, USA; keikobayashi0428@gmail.com (K.K.); liucm3@vcu.edu (C.L.); rjonas@childrensnational.org (R.A.J.); 2Children’s National Heart Institute, Children’s National Hospital, Washington, DC 20010, USA; 3School of Medicine, Virginia Commonwealth University, Richmond, VA 23298, USA; 4School of Medicine and Health Science, George Washington University, Washington, DC 20052, USA

**Keywords:** congenital heart disease, neuroprotection, brain, cardiac anomalies, neurological deficit

## Abstract

Neurological deficits are a serious and common sequelae of congenital heart disease (CHD). While their underlying mechanisms have not been fully characterized, their manifestations are well-known and understood to persist through adulthood. Development of therapies to address or prevent these deficits are critical to attenuate future morbidity and improve quality of life. In this review, we aim to summarize the current status of neuroprotective therapy in CHD. Through an exploration of present research in the pre-operative, intra-operative, and post-operative phases of patient management, we will describe existing clinical and bench efforts as well as current endeavors underway within this research area.

## 1. Introduction

Congenital Heart Disease (CHD) is the leading cause of death in infancy out of all known birth defects, occurring in 8 out of 1000 live births [1,2].However, the mortality of CHD has trended downwards as a result of improved diagnostic modalities, better surgical techniques, and advances in intensive care [3,4]. Now, most children affected by CHD will reach adulthood evident by the over 50% increase in adult prevalence rates from the years 2000 to 2010 [3]. This increase in survival rates has led to a corresponding shift in research focus from short-term survival to long-term morbidity and quality of life outcomes [5,6].

Neurodevelopmental deficits are a common and crucial sequela in patients with critical CHD, contributing to long-term morbidity that starts in infancy and persists into adulthood [3,6,7,8,9,10]. The prevalence and severity of such deficits are correlated with greater complexity of CHD and are estimated to reach an incidence of approximately 50% in infants [6,11]. Specifically, manifestations of neurodevelopmental deficits include impaired cognition, gross and fine motor skills, social interaction, communication, executive function as well as impulsive behavior and inattention [6,12,13,14] as presented from following clinical studies.

### 1.1. Boston Circulatory Arrest Trial

Longitudinal studies following patients with CHD throughout childhood have delivered valuable information about neurodevelopment. The Boston Circulatory Arrest Trial was a single-center longitudinal study that has followed a cohort of patients with dextrotransposition of the great arteries (d-TGA) who underwent the arterial switch operation (NCT03073122) [10,15,16]. In their patients’ infancy, Bellinger et al. reported numerous neurodevelopmental challenges including lower than normal scores on the Bayley Scales of Infant Development and substandard expressive language capabilities [17]. Childhood years continued this pattern of abnormal neurodevelopment evident in observations of lower intelligence quotient (IQ), worse mathematics and reading scores, poorer memory scores, attenuated visual-spatial skills, increased prevalence of behavioral problems, and immature motor function. Importantly noted was the finding that over a third of the cohort had received remedial services in school [13,14]. Age 16 follow-up demonstrated similar findings with persistent neuro-deficits and impaired psychosocial, physical, and emotional health [10,18]. Another study conducted at Boston Children’s Hospital observed similar neurodevelopmental outcomes in a cohort of 156 patients who had undergone Fontan procedures for single ventricle pathophysiology. Lower IQ scores, individual achievement, memory, executive function as well as an increased incidence of attention deficit hyperactivity disorder (ADHD), anxiety disorders, and MRI abnormalities paralleled the d-TGA cohort’s findings [19,20].

### 1.2. Single Ventricle Reconstruction Trial

Results consistent with these were also observed in the Single Ventricle Reconstruction (SVR) trial (NCT02692443). Recently published 6-year outcome data from the SVR trial revealed impairments in adaptive behavior, quality of life, functional status, and adaptive skills within a cohort of newborns with single right ventricular anomalies [21,22]. The incidence of these deficits surpasses that of the normal population with mild and severe motor disabilities occurring at 6 and 11 times the normal incidence rate respectively while the odds ratio for parent-reported executive function impairment has been reported to be 4.37 when compared to normative controls [23,24]. These reports are supported by several other studies describing similar results for cognitive, motor, and behavioral aspects of neurodevelopment [25,26,27,28,29,30]. Indeed, these deficits are common and are not specific to a single CHD subtype.

### 1.3. Risk Factors Associated with Neurological Deficits in Children with CHD

The causes of neurodevelopmental deficits associated with CHD are cumulative, multifactorial, and synergistic (Figure 1) [5]. Genetic components are unquestionably a broad underlying factor in the development of CHD and also a key contributor to a patient’s vulnerability to neurologic injury [5,31,32]. Additionally in the in utero phase, chronic fetal hypoxia resulting from abnormal physiology and altered regulatory pathways is largely responsible for immature brain development [33,34]. Establishment of a link between cerebral hypoxia and reduced brain volumes as well as poorer neurodevelopmental outcomes in CHD patients has been a crucial element in the push to develop pre-operative neuroprotective treatment aimed at addressing insufficient oxygen delivery to the fetus (Figure 2) [34,35].

Approximately 25% of neonates with CHD will require corrective cardiac surgery within their first year of life and will be at risk for neurological injury associated with cardiac surgery particularly when cardiopulmonary bypass (CPB) is required [36,37,38]. CPB causes systemic inflammatory response syndrome, ischemia-reperfusion, and reoxygenation injury as well as subsequent microglia mediated inflammation and enhanced oxidative stress, all of which contribute to the high incidence of white matter injury (WMI) observed in neonates and infants with CHD after surgery [39,40]. Although yet to be clearly defined, the mechanisms behind WMI are potential targets for neuroprotective therapy.

Post-operatively, factors contributing to neurodevelopmental delay are more broad in nature. Environment, socioeconomic status, and incidence of infections are factors that influence the maturation and growth for all infants, not only those with CHD [5,41,42,43,44]. Nonetheless, their significance in a patient population already at risk of neurodevelopmental deficits is elevated and must be explored to determine optimal methods of promoting healthy growth.

The gravity and prevalence of neurodevelopmental deficits within the CHD population necessitate therapies at each phase of patient management to prevent neurological injury and to promote neurodevelopment. Here, we review recent studies that illustrate the current status of neuroprotective treatment within CHD through analysis of existing strategies within the maternal, peri-operative, and post-operative stages of patient care (Table 1, Figure 3).

## 2. Maternal Neuroprotection—Clinical Studies

### 2.1. Maternal Hyperoxia

Chronic hypoxia is a key disruptor of fetal neurodevelopment in CHD. Recently, Lawrence et al. elucidated the role of chronic hypoxia in neurological injury by employing a fetal sheep model. They showed that chronic hypoxia in their artificial womb model reduced neuronal density and impaired myelination in the fetus resembling similar findings observed in neonates with CHD [45]. These neurological insults may be attenuated by supplementing oxygen supply. Previous studies have investigated the possibility of increasing oxygen levels in the fetus through maternal hyperoxygenation and have confirmed its feasibility through observed resultant increases in fetal oxygenation [46,47]. One clinical study in Canada has recently been initiated to investigate the effects of supplemental maternal oxygen during the 3rd trimester in mothers whose fetuses have been identified as having some form of CHD (https://clinicaltrials.gov/ ct2/show/NCT03944837, accessed on 3 November 2021). 10 to 15 L/min of oxygen by mask is given which briefly increases fetal oxygen levels to those reached in the newborn with spontaneous breathing. Although neurodevelopment is not directly defined as an endpoint, outcome measures of cerebral oxygen delivery, cerebral oxygen consumption, oximetry of major vessels, fetal brain volumetry, and brain size will provide valuable information about the physiological effects of maternal hyperoxygenation.

### 2.2. Progesterone

Although its exact cellular mechanism is still unknown, several animal studies have suggested that progesterone and its metabolites have a neuroprotective role in the developing brain [48]. In an acute hypoxia sheep model, allopregnanolone, a metabolite of progesterone through the enzymatic action of 5α-reductase, was found to be upregulated and correlated with an observed neuroprotective response after introduction of acute brain hypoxia [49]. A separate experiment confirmed this observation by observing a subsequent increase in neurological injury after inhibition of 5α-reductase via finasteride [50]. In addition, the same group also showed no increase in the fetal sheep brain allopregnanolone concentration after 20 days of chronic hypoxia created with a model of placental insufficiency in pregnant sheep, indicating a possible beneficial effect of allopregnanolone usage even in a chronic hypoxic state [51]. Investigation of WMI and brain immaturity in a neonatal rat model of chronic hypoxia, applied to mimic infants with cyanotic CHD, yielded similarly positive results. Notably, an increase in brain weights and greater motor and coordination abilities were observed in the progesterone treated group compared to controls [48]. Furthermore, progesterone has been implicated in oligodendrocyte maturation, oligodendrocyte progenitor cell proliferation, and myelination promotion through direct interaction with progesterone receptors [52]. As these effects of progesterone address cellular mechanisms of neurological injury observed in CHD, clinical translation of these studies has been pursued. Currently, a randomized phase 2 trial is underway in the USA to evaluate 3rd trimester Progesterone administration to a mother carrying a fetus with CHD (https://clinicaltrials.gov/ct2/show/NCT02133573, accessed on 3 November 2021). With endpoints specifically studying neurodevelopmental scores and brain maturation via MRI, a more thorough understanding of progesterone’s effects on neuroprotection will be achieved at its conclusion.

### 2.3. Topiramate Prophylaxis

Topiramate is an α-amino-3-hydroxy-5-methyl-4-isoxazolepropionic acid (AMPA) receptor antagonist approved for the treatment of partial and generalized epileptic seizures. Recently it has been shown that Topiramate offers a potential therapy for hypoxic-ischemic encephalopathy by reducing underlying over-excitatory mechanisms of neural injury through increased gamma butyric acid activity (GABA) and decreased glutamate activity to address excessive glutamate production in the brain [53]. Combining study results that have demonstrated its neuroprotective capacity and ability to preserve oligodendrocytes in traumatic brain injury (TBI), stroke, epilepsy, and periventricular leukomalacia models with reports of improved long term cognition, memory, and attenuated WMI when used in tandem with therapeutic hypothermia has propelled Topiramate to the clinical trial phase [54,55,56,57]. Filippi et al. recently reported the results of a phase 2 multi-centre randomized control trial evaluating topiramate in neonates with hypoxic ischemic encephalopathy (HIE) [58]. In their study, they observed a non-significant downward trend in rates of epilepsy development in the topiramate treated group compared to controls but no significant difference in mortality and severe neurodevelopmental disability. However, this study was limited by its small number in the original cohort and the one-third reduction in sample size due to incorrect randomization. A separate study completed in 2017 studying the effects of topiramate prophylaxis in infants with CHD undergoing cardiac surgery will provide more information about the applicability of topiramate specifically in CHD as well as the future direction of this intervention (https://clinicaltrials.gov/ct2/show/NCT01426542, accessed on 3 November 2021).

## 3. Maternal Neuroprotection—Preclinical Studies for Future Translation

### Tetrahydrobiopterin

Tetrahydrobiopterin (BH4) is key component of neuronal NO synthase function. During cerebral hypoxia, BH4 levels are depleted, leading to neuronal NO synthase uncoupling and a subsequent shift in production from NO to peryoxynitrite [59,60]. The ensuing damage generated by peryoxynitrite’s interaction with DNA and the increased production of reactive oxygen species escalates apoptotic and necrotic cell death, ultimately culminating in neurological injury [61]. As such, addressing the exhaustion of BH4 could be advantageous in protecting against WMI injury in the context of hypoxia. A recent study from Romanowicz et al. tested this hypothesis in a hypoxic rat model mimicking 3rd trimester human development [62]. In their experiment, they revealed that BH4 treatment prevented chronic hypoxia-associated delayed myelination and diminished mature oligodendrocyte populations while broadly decreasing apoptosis, although not specifically in oligodendrocytes. Crucially, BH4 treated mice showed protected motor-sensory coordination compared to control mice. These data are promising, suggesting possible auxiliary use of an FDA-approved agent for prenatal neuroprotective therapy.

## 4. Peri-Operative Neuroprotection—Clinical Studies

### 4.1. Corticosteroids

Corticosteroids have been used in cardiac surgery for more than 60 years. Mechanisms of action include modulation of CPB associated systemic inflammatory response [63] and possible counteraction of surgery-related relative adrenal insufficiency [64]. Several randomized controlled studies have shown that perioperative steroids reduce post-CPB inflammatory markers [65,66,67] However, the intraoperative use of corticosteroids in corrective congenital cardiac surgery is a highly debated topic [68]. Notwithstanding the numerous trials conducted, the long-term neurological effects of corticosteroids have yet to be studied [69,70]. Recently, a secondary analysis of a two-center, double-blind, randomized, placebo-controlled trial by Zyblewski et al. finally offered a glimpse into the neuroprotective effects of methylprednisolone [71]. They found that no significant difference was present in 12-month Bayley-III Developmental scores and post-operative brain injury biomarker levels between patients who received intra-operative methylprednisolone or placebo. Currently a phase 3 trial investigating the safety and efficacy of methylprednisolone in infants undergoing heart surgery with CPB is ongoing. The trial, when completed, will be one of the largest trials ever and will hopefully provide some clear answers regarding the use of corticosteroids for congenital heart surgery. (https://clinicaltrials.gov/ct2/show/NCT03229538, accessed on 3 November 2021).

### 4.2. Remote Ischemic Preconditioning

Ischemic preconditioning is a protective strategy against ischemia-reperfusion brain injury and is defined as the introduction of transient periods of ischemia prior to a prolonged period of ischemia such as hypothermic circulatory arrest [72]. The concept of remote ischemic preconditioning (RIPC) is similar but is distinguished by its ability to protect organs separate from the organ being exposed to transient ischemia [72]. Several animal studies have demonstrated the effects of RIPC in the protection of cerebral injury from stroke and hypothermic circulatory arrest [72,73,74]. As a result numerous clinical trials have been launched to translate these results to the clinical setting. Gaynor et al. recently published a randomized control trial evaluating RIPC and sham interventions in neonates before CPB commencement [75]. They identified no significant difference in WMI incidence or change in WMI volume change between groups. Another trial reported a similar lack of difference in neurodevelopmental outcomes when measuring post-operative neuron-specific enolase (NSE) and S100B levels [76]. However, they did observe cardioprotective effects with a decrease in post-operative Troponin1 and Creatine Kinase-MB levels in patients who received RIPC. Although these studies provide a greater understanding of the immediate post-operative effects of RIPC, a clinical trial underway in the United States will elaborate on RIPC in a longer term setting by being the first trial to report 12-month Bayley Developmental Scores in patients with CHD (https://clinicaltrials.gov/ct2/show/NCT01835392, accessed on 3 November 2021).

### 4.3. Mesenchymal Stromal Cells

Mesenchymal stromal cells (MSC) are a potential therapeutic agent for hypoxic brain injury due to their immunomodulatory and regenerative properties [77]. Various studies have shown that MSCs accelerate white matter remyelination through the activation of endogenous oligodendrocyte progenitors, promote neurogenesis from subventricular zone neural progenitors, and regulate microglia activation after hypoxic ischemic brain insults [78,79]. A recent laboratory study showed that MSC delivery through CPB has the potential to mitigate effects of CPB on neural stem/progenitor cells and to promote migration of neuroblasts in the subventricular zone [79]. In that study MSCs were delivered intraoperatively rather than preoperatively because cardiac surgery’s use of CPB provides arterial access which prevents loss of MSCs in the lungs observed during venous injection. The practicality of this method of delivery was recently assessed in an ex vivo CPB model. The study found that MSC delivery does not interfere with oxygenator function and does not elicit an immunogenic response in the host [80]. Together these findings have culminated in the initiation of a phase 1 prospective, open-label, single-center study which will determine the safety and feasibility of delivering MSCs in patients with CHD as well as any neurodevelopmental, neuroimaging, and postoperative inflammatory differences (https://clinicaltrials.gov/ct2/show/NCT04236479, accessed on 3 November 2021).

### 4.4. Erythropoietin

Erythropoietin (EPO) has been investigated as a potential neuroprotective pharmacologic intervention in neonates [81]. Although a previous meta-analysis of four randomized trials by Hendrik et al. demonstrated EPO reduced the risk of neurodevelopmental impairment as measured by the Mental Developmental Index score at an age of 18 to 24 months [82], a recent randomized trial by Juul et al. showed high-dose EPO administration to extremely preterm infants did not result in a lower risk of severe neurodevelopmental impairment or death at 2 years of age [83]. In addition a clinical trial specifically focusing on intraoperative and peri-operative effects of EPO administration failed to show significant neurodevelopment outcome improvement at 1 year of life in neonates who underwent surgery for D-transposition of the great vessels, hypoplastic left heart syndrome, or aortic arch reconstruction, albeit with a small sample size [84]. Nevertheless, this study showed that the studied doses of EPO were safe in the neonate and will require future appropriately powered studies to elucidate the true effects of EPO on neurodevelopment in CHD patients following neonatal corrective surgery.

### 4.5. Dexmedetomidine

Prolonged and repeated use of anesthetics results in an important risk of developmental neurotoxicity in patients with CHD [85,86]. Dexmedetomidine, an α2 adrenergic receptor agonist, has been explored in several studies to test its neuroprotective properties in a CHD population [87,88,89]. Results have demonstrated an association between Dexmedetomidine administration and reduced levels of biomarkers of neurological injury, NES and S-100B, and have led to the initiation of ongoing clinical trial in CHD patient populations (https://clinicaltrials.gov/ct2/show/NCT02492269, accessed on 3 November 2021). Additionally other investigations into the utilization of Sevoflurane and specific anesthetic strategies have joined Dexmedetomidine in the clinical trial phase and may produce valuable insights as to how to better promote preservation of neurologic function in CHD patients undergoing cardiac surgery (https://clinicaltrials.gov/ct2/show/NCT03882788, accessed on 3 November 2021, https://clinicaltrials.gov/ct2/show/NCT02492269, accessed on 3 November 2021, https://clinicaltrials.gov/ct2/show/NCT03366597, accessed on 3 November 2021, https://clinicaltrials.gov /ct2/show/NCT04484922, accessed on 3 November 2021). Despite the safety and effectiveness of dexmedetomidine, it is known to have cardiac toxicities including bradycardia and hypotension. Dose adjustments should be considered especially with neonates who have reduced clearance [90]

### 4.6. Tight Glycemic Control

Neurodevelopment has been assessed within the context of perioperative glycemic control.

Studies linking neonatal hypoglycemia with adverse neurodevelopmental outcomes and hyperglycemia with microglial activation and neuronal damage to the hippocampal and frontal cortex have established the importance of maintaining a euglycemic state in neonates [91,92]. This has led to the idea that there may be a role for tight glycemic control in the care of neonates with CHD. In 2016, Sadhwani et al. reported the results of a two-center, prospective, randomized trial assessing infant neurodevelopment in CHD patients randomized to tight glycemic control or standard care post-operatively, ultimately finding no differences in neurodevelopmental scores at 1 year of age between groups [93]. Furthermore a study and its post hoc analysis by Agus et al. revealed that while infections were reduced in infants greater than 60 days old within the tight glycemic control group, tight glycemic control did not lead to a difference in mortality, length of stay, or overall infection rates [94]

### 4.7. Allopurinol

Xanthine oxidase is a potent source of free oxygen radicals, especially superoxide. Allopurinol is a xanthine oxidase inhibitor, which reduces the production of oxygen radicals [95]. In a rodent model of acute hypoxia-ischemic brain injury, allopurinol was shown to decrease brain injury through its antioxidant properties grounded in its ability to chelate unbound iron and scavenge free hydroxyl radicals [96]. Several clinical studies demonstrated a possible neuroprotective effect in neonates with HIE. Gunes et al. later found in a small randomized clinical trial of neonates with HIE that allopurinol treatment was associated with better developmental outcomes compared with placebo at one year of age [97]. Four to five-year follow ups of a separate trial of asphyxiated patients broadly showed no difference in mortality or adverse developmental outcomes [98]. However, sub-group analysis revealed a decrease in severe adverse outcomes defined as mortality or severe disability in neonates with moderate HIE who received allopurinol. While the effects of allopurinol in the context of CHD are unknown, these studies have demonstrated potential benefits in animal models and neonates with encephalopathy. A prospective randomized phase 3 trial currently taking place in the Netherlands is studying the perioperative and postnatal administration of allopurinol in CHD patients (https://clinicaltrials.gov/ct2/show/NCT04217421, accessed on 3 November 2021).

## 5. Peri-Operative Neuroprotection—Preclinical Studies for Future Translation

### 5.1. Inhaled H_2_

H_2_ gas is a potential agent capable of addressing CPB-associated neurological injury. It is known that the generation of superoxide anions and subsequent production of toxic hydroxyl radicals occurs in neurodegenerative and neuroinflammatory disorders [99]. Without an endogenous detoxification mechanism to neutralize these hydroxyl radicals, humans are susceptible to the cellular injury and increased apoptosis that take place when reactive oxygen species (ROS) interact with cellular components [100]. H_2_ gas may be able to fill this deficit through its role as a reducing agent [101]. Indeed, improved neurological scores, myocardial function, and survival have been associated with H_2_ gas treatment in a study comparing H_2_ treated rodents to rodents treated with only therapeutic hypothermia after 5 min of asphyxia induced cardiac arrest [102]. These results paired with CPB’s association with cerebral hypoxia have led to further study in CPB models. Recently Cole et al. studied continuous pre- and post-operative administration of 2.4% H_2_ gas in a neonatal swine model of CPB-induced cerebral hypoxic-ischemic injury [103]. They found that H_2_ gas administration led to greater rates of neurologically intact survival, improved neurologic deficit scores, and lower volumes of WMI both by MRI and histological analysis. The positive results of this study have led the authors to suggest possible applications of H_2_ in CHD heart surgery as well as in the settings of extracorporeal membrane oxygenation, myocardial infarction, and stroke.

### 5.2. Inhaled Nitric Oxide

In addition to CPB, deep hypothermic circulatory arrest (DHCA) potentiates development of neurological injury in CHD patients. One of the causes is disruption of nitric oxide (NO) regulated processes including ischemia/reperfusion, cerebral blood flow autoregulation, and microglial activation [104]. Inhaled NO (iNO) use during pediatric CPB has become popular and several trials have been reported [105,106]. These studies demonstrate improvement of post-operative outcomes determined by the anti-inflammatory effects of NO but are not specific to neuronal protection. The use of iNO requires further studies.

iNO exerts its neuroprotective effects by preserving cerebral autoregulation, an important factor in preserving oxygen delivery to the brain. This mechanism was better defined in a porcine model of TBI which showed that protection of cerebral autoregulation and subsequent decrease in hippocampal injury can be attributed to inhibition of Endothelin 1 (ET-1) and Extracellular Signal-Regulated Kinase (ERK)/Mitogen Activated Protein Kinase (MAPK) and simultaneous IL-6 upregulation [107]. The translation of these findings to models of cardiac surgery is reasonable as TBI and cardiac surgery share sequelae of impaired cerebral autoregulation and neuronal damage. Kajimoto et al. recently investigated the anti-inflammatory effects of inhaled Nitric Oxide in porcine models specifically in the setting of heart surgery with DHCA [104]. The study showed that iNO treated pigs had less neuronal degeneration, smaller microglial cell body volume, longer dendrite process length, and a larger quantity of branch segments and terminal branch points. Altogether these findings are suggestive of decreased microglial activation and maintain consistency with previous studies of iNO’s neuroprotective mechanisms. These results from basic research will likely lead to future clinical studies to determine the effectiveness of iNO in mitigating the deleterious effects of CPB and cardiac surgery on neuronal development.

### 5.3. Whole Body Periodic Acceleration

The suggested neuroprotective effects of whole-body periodic acceleration (pGz) are also based on its potential to modulate NO regulated pathways. Defined as a rhythmic pattern of acceleration and deceleration along the head to foot axis akin to the motion generated when pushing a stroller back and forth, pGz has been hypothesized to activate endothelial Nitric Oxide Synthase (eNOS) pathways via pulsatile shear stress generation [108]. Upregulation of these pathways, in turn, promotes cardiac and neuroprotective effects through their anti-inflammatory and anti-apoptotic properties [109]. A porcine model was used to investigate pGz preconditioning as a strategy for neuroprotection in cardiac surgery and found expected activation of eNOS and increased activation of anti-apoptotic p-Akt/Akt and Bcl/Bax signaling [110].

### 5.4. Triptolide

Triptolide is an extract of *Tripterygium wilfordii,* a plant commonly used in traditional Chinese medicine [111]. Known for its anti-inflammatory and immunosuppressive activity, Triptolide is a prospective therapeutic for addressing upregulated inflammatory pathways associated with CPB. Previous in vitro studies modeling traumatic brain injury, cerebral ischemia/reperfusion injury, and stroke have confirmed the presence of anti-inflammatory effects driven by suppressed activation of the NF-kB and p38MAPK pathways [112,113,114]. Importantly, these studies showed attenuated neurological deficits to be associated with Triptolide treatment [113]. A more recent experimental study applied these results to a 12–14 week rat model of CPB with DHCA and concluded that Triptolide treatment led to decreased levels of TNF a, IL1b, IL6, malondialdehyde, and ROS while increasing glutathione and superoxide dismutase levels [115]. Moreover mitigated microglia activation, NF-kB activity inhibition, and upregulation of the NRF2 pathway were observed, further emphasizing Triptolide’s anti-inflammatory and neuroprotective qualities. Neurodevelopmental endpoints in the same study were found to be positively impacted by Triptolide administration evident in improved spatial learning, memory, and anxiety-like behaviors. Given that the model used in this study was an adult rat, the results should be interpreted carefully for neonates. Further study in a neonate model is needed.

### 5.5. Minocycline

Minocycline is a tetracycline antibiotic commonly used for chronic recurrent bacterial infections [116]. Lately, it’s anti-inflammatory and neuroprotective properties have been highlighted in several animal models. Drabek et al. showed in their rat model that minocycline significantly attenuated brain tumor necrosis factor alpha, a principal mediator of neuroinflammation, after CPB followed by DHCA [117]. Aida et al. demonstrated significant attenuation of markers for hypoxia and apoptosis in cells from the hippocampus of 4-week-old piglets with 90 min of CPB followed by minocycline administration [118]. Results from these animal studies may provide sufficient rationale to perform clinical studies in the near future. Although developments in the understanding of minocycline’s neuroprotective capacity suggest potential applications for attenuation of neurologic injury, it should be remembered that in neonates, tetracycline can cause stunting of bone growth and bilirubin-induced brain damage [119].

## 6. Post-Operative Neuroprotection—Clinical Studies

### 6.1. Triiodothyronine

Thyroid hormones are a critical component of neurodevelopment in the post-natal period. In particular, the processes of cerebellar neurogenesis, gliogenesis, and myelogenesis rely on adequate Triiodothyronine levels while hypothyroxinemia has been observed to be associated with developmental delay [120]. Even transient congenital hypothyroidism in neonates has been reported to be associated with lower IQ at 7–8 years of age [121].

Reductions in thyroid hormone levels have been reported postoperatively after pediatric and adult cardiac surgery [122,123]. Bettendorf et al. attempted to raise thyroid levels in a cohort of 40 children undergoing cardiac surgery with triiodothyronine treatment and was able to verify that triiodothyronine is efficacious in elevating triiodothyronine plasma levels while improving myocardial function and reducing postoperative intensive care [124]. The long-term neuroprotective effects of triiodothyronine supplementation were assessed for the first time in the same cohort 10 years later. In the follow-up study, no significant differences were observed in IQ, gross and fine motor skills, and executive function between the triiodothyronine treated and control groups [125].

### 6.2. Standardized Exercise Program

The numerous benefits of physical activity are well established [126]. Importantly the impact of physical activity has also been shown to extend to aspects of cognition and neurodevelopment. Several studies in adults have demonstrated evidence suggestive of relationships between physical activity and hippocampal volume preservation as well as increased brain volume [127,128,129,130]. One randomized control trial even demonstrated a 2% increase in hippocampal volume corresponding to better spatial memory in response to a physical activity regimen [131]. In preadolescent children, improved executive function, attention, and academic performance have also been found to be associated with physical activity [132].

Similar results have been reported in CHD. Dulfer et al. randomized patients 10 to 25 years old with Tetralogy of Fallot or single ventricle anomalies into a 3-month standardized exercise program [133]. Aside from demonstrating better quality of life measures in the exercise group, the authors also observed positive effects in self-reported cognitive functioning and parent-reported social functioning. In addition, an ongoing clinical trial at Columbia University will attempt to show similar findings in a younger patient group through IQ, cognitive function, and adaptive behavior evaluations at 24 months (https://clinicaltrials.gov/ct2/show/NCT02542683, accessed on 3 November 2021).

### 6.3. Physical Therapy

The effects of physical therapy in the context of CHD have been rarely described. However, an observational cohort study in 2021 looking at gross motor development in children with CHD found increased Bayley-III scores during 12–24 month up assessments in patients who received regular physical therapy compared to those who received either no physical therapy or occasional physical therapy [134]. A recent study reported that the majority of a CHD cohort did not meet guidelines for physical fitness and had not received physical therapy 2 years post-operatively. Further investigation should be conducted to determine if participation in physical therapy by patients with CHD will promote motor skill development [135].

### 6.4. Cogmed Working Memory Training

The Cogmed Working Memory Training is a 5 week computerized program designed to enhance executive function, organization, and attention [136]. While its use has been shown to impart these benefits in children and adolescents with ADHD, learning disabilities as well as those who were born pre-term, its effects have yet to be demonstrated in patients with CHD [137,138,139]. Recently a clinical trial studied the efficacy of Cogmed Working Memory Training in a group of 13–16 year old adolescents. This trial specifically found that the training improved inhibitory control, attention, planning, and organizational skills immediately after training and at 3-month follow-up [136]. Furthermore the group assigned to the training demonstrated higher social responsiveness and communication scores. Although components of working memory and processing speed did not improve as previously reported in other patient populations, the Cogmed Working Memory Training had an overall positive impact on the neurological function of adolescents with CHD. A trial recently completed in 2020 is anticipated to show similar improvements specifically in patients with CHD in the age range of 7–12 years https://clinicaltrials.gov/ct2/show/NCT03023644, accessed on 3 November 2021).

### 6.5. Early Stimulation

The environment of the patient with CHD is crucial for their neurodevelopment. Among known non-biological factors such as maternal education and deprivation, a cognitively stimulating environment is known to be a modifiable factor with the capability to overcome other factors contributing to deficits in neurodevelopment [140,141,142,143]. Bonthrone et al. retrospectively assessed the level of cognitive stimulation of children with CHD by their parents and found that higher parent cognitive stimulation scores were associated with higher 22 month language and cognitive abilities [144]. A research group at the Instituto de Cardiologia de Rio Grande do Sul is investigating the effects of parent-administered early stimulation programs for children with CHD in a randomized prospective clinical trial by using 3 and 6 month neurodevelopmental scores as their primary endpoints (https://clinicaltrials.gov/ct2/show/NCT04152330, accessed on 3 November 2021).

## 7. Socioeconomic Status

We can’t disregard and close this review without talking about socioeconomic status (SES). SES is a measure of one’s overall status and position in society, which is well-recognized predictor of neurodevelopmental outcome in preterm children and may attenuate the effect of brain injury particularly on cognitive development [44]. Indeed, in many studies reporting shorter and longer-term neurodevelopmental outcomes, lower SES has been identified as an independent risk factor for worse outcomes in the CHD population [10,145,146,147]. Less maternal education has been associated with lower mental development index in infants with CHD [148]. Thus, particular attention must be given to neurodevelopmental care during the hospitalization and after discharge in children from disadvantaged families. Recent study reporting the relationship between a stimulating home environment and cognitive abilities in toddlers with CHD showed no relationship between outcome scores and SES, clinical factors, or brain injury severity at 22 months [144] However, the sample size is relatively small and future studies assessing the impact of home environment stimulation with larger samples are required.

## 8. Concluding Remarks and Future Perspectives

Neurodevelopmental deficits are common and important sequelae of CHD that are highly complex with cumulative, multifactorial, and synergistic etiologies. An important current limitation is that many clinical trials undertaken in the CHD population are based on clinical trials in other populations or on preclinical studies that target different mechanisms of injury. Therefore, the signaling pathways of the possible mechanisms and physiological events in children with CHD are not fully understood. While current therapies are being developed, continued collaboration and effort are needed to elucidate and dictate future areas of potential neuroprotective therapy in all stages of patient care. In addition, genetic contributions are now becoming a very important topic [31,32]. Although there is a need for better understanding of the impact of CHD-linked genes on brain development, the studies presented in this paper shed light on potential future therapeutic options.

## Figures and Tables

**Figure 1 children-08-01116-f001:**
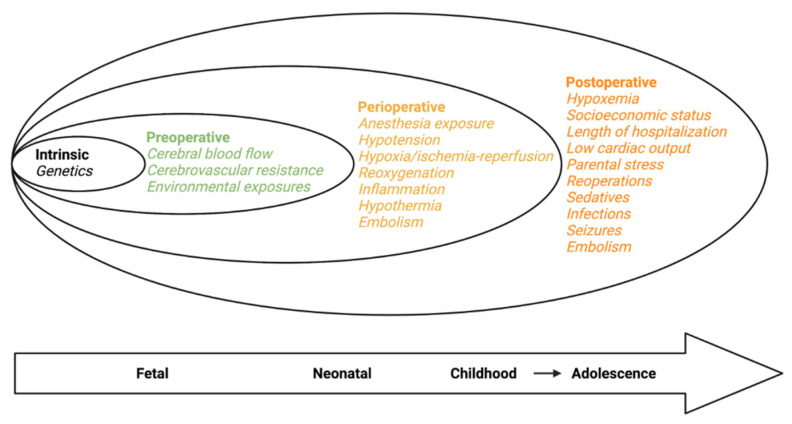
Risk factors associated with neurodevelopmental outcomes in congenital heart disease during progressive epochs of brain development. Adapted from Morton et al. Circ Res. 2017.

**Figure 2 children-08-01116-f002:**
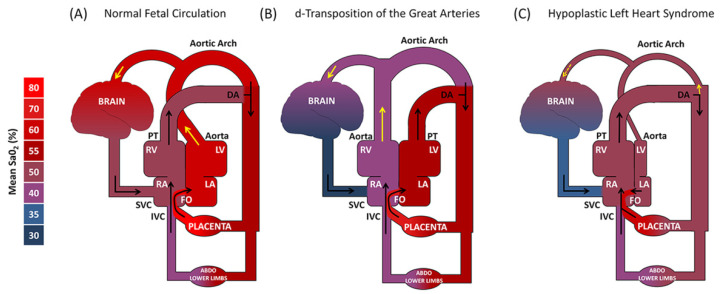
In utero cerebral hypoxemia in complex congenital heart defects. Figure shows fetal hemodynamics in representative examples of (**A**) Normal Fetal Circulation, (**B**) d-Transposition of the Great Arteries, and (**C**) Hypoplastic Left Heart Syndrome (Leonetti et al. Trends Neurosci 2019).

**Figure 3 children-08-01116-f003:**
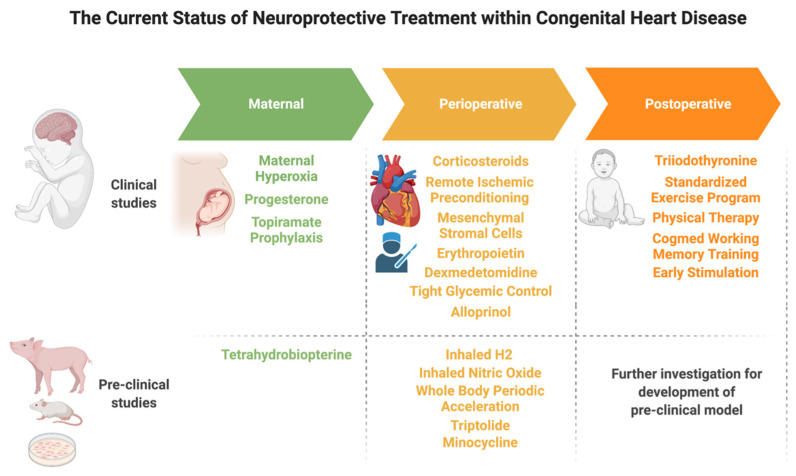
The current status of neuroprotective treatment for congenital heart disease.

**Table 1 children-08-01116-t001:** The current status of neuroprotective treatment for congenital heart disease. n/a, not applicable.

Stage of Patient Care	Type of Study	Treatment	References	Clinical Trial
Maternal	Clinical	Maternal Hyperoxia	[45,46,47]	https://clinicaltrials.gov/ct2/show/NCT03944837, accessed on 3 November 2021
		Progesterone	[48,49,50,51,52]	https://clinicaltrials.gov/ct2/show/NCT02133573, accessed on 3 November 2021
		Topiramate Prophylaxis	[53,54,55,56,57,58]	https://clinicaltrials.gov/ct2/show/NCT01426542, accessed on 3 November 2021
	Pre-clinical	Tetrahydrobiopterine	[59,60,61,62]	n/a
Perioperative	Clinical	Corticosteroids	[63,64,65,66,67,68,69,70,71]	https://clinicaltrials.gov/ct2/show/NCT03229538, accessed on 3 November 2021
		Remote Ischemic Preconditioning	[72,73,74,75,76]	https://clinicaltrials.gov/ct2/show/NCT01835392, accessed on 3 November 2021
		Mesenchymal Stromal Cells	[77,78,79,80]	https://clinicaltrials.gov/ct2/show/NCT04236479, accessed on 3 November 2021
		Erythropoietin	[81,82,83,84]	n/a
		Dexmedetomidine	[85,86,87,88,89,90]	https://clinicaltrials.gov/ct2/show/NCT02492269, accessed on 3 November 2021
				https://clinicaltrials.gov/ct2/show/NCT03882788, accessed on 3 November 2021
				https://clinicaltrials.gov/ct2/show/NCT02492269, accessed on 3 November 2021
				https://clinicaltrials.gov/ct2/show/NCT03366597, accessed on 3 November 2021
				https://clinicaltrials.gov/ct2/show/NCT04484922, accessed on 3 November 2021
		Tight Glycemic Control	[91,92,93,94]	n/a
		Alloprinol	[95,96,97,98]	https://clinicaltrials.gov/ct2/show/NCT04217421, accessed on 3 November 2021
	Pre-clinical	Inhaled H2	[99,100,101,102,103]	n/a
		Inhaled Nitric Oxide	[104,105,106,107]	n/a
		Whole Body Periodic Acceleration	[108,109,110]	n/a
		Triptoide	[111,112,113,114,115]	n/a
		Minocycline	[116,117,118,119]	n/a
Postoperative	Clinical	Triiodothyronine	[120,121,122,123,124,125]	n/a
		Standardized Exercise Program	[126,127,128,129,130,131,132,133]	https://clinicaltrials.gov/ct2/show/NCT02542683, accessed on 3 November 2021
		Physical Therapy	[134,135]	n/a
		Cogmed Working Memory Training	[136,137,138,139]	https://clinicaltrials.gov/ct2/show/NCT03023644, accessed on 3 November 2021
		Early Stimulation	[140,141,142,143]	https://clinicaltrials.gov/ct2/show/NCT04152330, accessed on 3 November 2021

## Data Availability

Not applicable.

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
