# Peer review of "The Current Status of Neuroprotection in Congenital Heart Disease"

_children, 2021, doi:10.3390/children8121116_

Round 1

Reviewer 1 Report

Kobayashi and colleagues have provided an excellent comprehensive review of the current state of neuroprotection in congenital heart disease.  This review spans the current considerations and evidence in neuroprotection throughout the perioperative period.  The writing is concise and clear.  This work should and will be considered a reference for anyone caring for children undergoing congenital heart surgery.

My only feedback is regarding the topic of socioeconomic status (SES) and its impact on neurologic outcomes.  I applaud the authors for making mention of SES in figure 1; however, I was disappointed to see that there was not a section dedicated to this topic.  Certainly, SES and neurologic outcomes in congenital cardiology may not be the most scientifically rigorous topic; however, I believe the authors can make an impact on this very important topic by dedicating just one paragraph toward the end of the manuscript.  Nevertheless, a search of the literature returns a number of relevant articles on this topic that could/should be referenced and summarized in this review.

Author Response

Kobayashi and colleagues have provided an excellent comprehensive review of the current state of neuroprotection in congenital heart disease.  This review spans the current considerations and evidence in neuroprotection throughout the perioperative period.  The writing is concise and clear.  This work should and will be considered a reference for anyone caring for children undergoing congenital heart surgery.

RESPONSE: Thank you for the kind words.

My only feedback is regarding the topic of socioeconomic status (SES) and its impact on neurologic outcomes.  I applaud the authors for making mention of SES in figure 1; however, I was disappointed to see that there was not a section dedicated to this topic.  Certainly, SES and neurologic outcomes in congenital cardiology may not be the most scientifically rigorous topic; however, I believe the authors can make an impact on this very important topic by dedicating just one paragraph toward the end of the manuscript.  Nevertheless, a search of the literature returns a number of relevant articles on this topic that could/should be referenced and summarized in this review.

RESPONSE: Thank you for the comments. We totally agree that SES is very important topic in neurologic outcomes in children with CHD. As suggested, new section about SES has been added in the revised version of our manuscript (page 12, section 7).

Reviewer 2 Report

Kobayashi et al give a good summary of the current status of neuroprotection therapy in the pre-operative, intra-operative, and post-operative phases of congenital heart disease patient management, the manuscript is well written, however, there are still some flaws that need further improvement in some aspects.

  1. Tables are recommended to vividly summarize neuroprotective treatment within the maternal, peri-operative, and post-operative stages.
  2. It is best to provide maps of the signaling pathways of the possible mechanisms of various neuroprotective strategies at different stages of care.
  3. Figure legends should be provided to briefly describe all figures (Figure 1-3).
  4. Introduction is too long, and should be shortened or divided into small paragraphs.
  5. Please kindly provide ClinicalTrials.gov Identifier or website for The Boston Circulatory Arrest Trial, Single Ventricle Reconstruction (SVR) trial.

Author Response

Kobayashi et al give a good summary of the current status of neuroprotection therapy in the pre-operative, intra-operative, and post-operative phases of congenital heart disease patient management, the manuscript is well written, however, there are still some flaws that need further improvement in some aspects.

1. Tables are recommended to vividly summarize neuroprotective treatment within the maternal, peri-operative, and post-operative stages.

RESPONSE: As recommended, we have added a summarized table which summarizes neuroprotective treatment within the maternal, peri-operative, and post-operative stages (see Table 1)

2. It is best to provide maps of the signaling pathways of the possible mechanisms of various neuroprotective strategies at different stages of care.

RESPONSE: We agree that we need to understand the mechanism for each treatment.  An important current limitation is that many clinical trials undertaken in the CHD population are based on clinical trials in other populations or on preclinical studies that target different mechanisms of injury. Therefore the signaling pathways of the possible mechanisms and physiological events in children with CHD are not fully understood. We understand the need to elucidate mechanism for further improvement. The limitation discussed above is now included in revised “Concluding remarks and future perspectives” (page 13 line 512-13).

3. Figure legends should be provided to briefly describe all figures (Figure 1-3).

RESPONSE: Thank you for your comment. Figure legends have been included in our revised manuscript.

4. Introduction is too long, and should be shortened or divided into small paragraphs.

RESPONSE: Thank you for this suggestion. Introduction is now subdivided into small paragraphs, and titles are added in each paragraph (page 2).

5. Please kindly provide ClinicalTrials.gov Identifier or website for The Boston Circulatory Arrest Trial, Single Ventricle Reconstruction (SVR) trial.

RESPONSE: ClinicalTrials.gov Identifier for the latest ‘The Boston Circulatory Arrest Trial’ is NCT03073122. ClinicalTrials.gov Identifier for the latest ‘SVR trial’ is NCT02692443. The numbers are added in our revised manuscript (page 2 line 52 and line 70).

Round 2

Reviewer 2 Report

Thank you for your rapid revision. Here are some minor suggestions:

  1. N/a should be annotated as Not Applicable in Figure legends in Table 1.
  2. Regarding perioperative stage of patient care, 'https://clinicaltrials.gov/ct2/show/NCT04484922' was written in duplicate.
  3. 'Standardized exercise program' should be capitalized as 'Standardized Exercise Program' in Table 1.

Author Response

Response to reviewer 2:

Thank you for your rapid revision. Here are some minor suggestions:

  1. N/a should be annotated as Not Applicable in Figure legends in Table 1.

RESPONSE: As recommended, we have added the annotation in Figure 1 legend.

  1. Regarding perioperative stage of patient care, 'https://clinicaltrials.gov/ct2/show/NCT04484922' was written in duplicate.

RESPONSE: Thank you for this suggestion. We have deleted the duplicate from Table 1 and page 8, line 290.

  1. 'Standardized exercise program' should be capitalized as 'Standardized Exercise Program' in Table 1.

RESPONSE: Thank you for this suggestion. We have capitalized the term in Table 1. Term is also changed in Figure 3 and graphical abstract accordingly